# Ultrasonic Measurement of Stress in SLM 316L Stainless Steel Forming Parts Manufactured Using Different Scanning Strategies

**DOI:** 10.3390/ma12172719

**Published:** 2019-08-25

**Authors:** Xiaoling Yan, Jincheng Pang, Yanlong Jing

**Affiliations:** College of Material Science and Mechanical Engineering, Beijing Technology and Business University, Beijing 102488, China

**Keywords:** SLM forming parts, additive manufacturing, scanning strategies, stress, critical refraction longitudinal wave

## Abstract

Selective Laser Melting (SLM) technology is a new kind of additive manufacturing technology developed in in the last decade. Measurement and control of stress in metal forming layer is the basic problem of SLM forming parts. Critical Refraction Longitudinal (LCR) wave method was used to measure stress. The acoustic-elastic formulas for measuring stresses in SLM 316L stainless steel forming parts manufactured using meander, stripe, and chess board scanning strategies, respectively, were established based on static load tensile test. The experimental results show that the acoustic time difference of LCR wave in SLM specimen manufactured with 316L stainless steel increases linearly with the increase of stress when the tensile stress is less than critical stress (372 MPa, 465 MPa, and 494 MPa). Due to the inhomogeneous deformation of the anisotropic SLM forming layer and the dimple-micropore aggregation fracture mechanism, the acousto-elastic curve fluctuates up and down along the irregular curve when the tensile stress is larger than critical stress. The results of corroboration experiments show that nondestructive measurement of stress in SLM forming specimen can be realized by using LCR wave method. The scanning strategy can significantly affect the tensile strength and yield strength of SLM forming specimen. The stresses were all in tension stress state at the edge of the specimens, whatever scanning strategy was used. Sub-area scanning and scanning sequence of alternate and intersect were adopted, which can effectively reduce the stress in the SLM forming specimen. The overall stress values of SLM forming specimen manufactured using chess board scanning strategy were smaller than that using meander and stripe strategies. The distribution of stress were more uniform.

## 1. Introduction

Selective Laser Melting (SLM) technology is a new kind of additive manufacturing technology developed in the last decade. The discrete layer profile information of three-dimensional model is used by SLM technology to control high-energy laser beam to melt metal powder layer by layer, thus to prepare metal parts. Compared with traditional manufacturing technology, SLM technology is theoretically not limited to complex geometries, and it can form functional parts with nearly 100% density of any complex structure. SLM technology [1,2,3] has been extended from conceptual mould design to mould design in aerospace, biomedical and automotive fields.

The single-point high-energy loading mode adopted by SLM technology makes the energy loading and forming time in sequence, and the temperature gradient is large and the cooling speed is fast. The heat dissipation speed and the constraint force formed during cooling contraction of metal powder melted at different positions in the same layer vary with the scanning path, result in uneven shrinkage of melted metal powders at different positions, which greatly increases the difficulty of controlling the forming quality. Especially when large size parts are formed, the scanning line is too long and the residual stress [3,4] accumulated in the layer is too large, which can easily cause parts to warp and crack, and even affect the powder laying process seriously, leading to the interruption of parts forming process. So, measurement and control of stress [5,6] in the metal forming layer is the basic problem of SLM forming parts. Meander scanning, stripe scanning, and chess board scanning are three basic scanning methods in SLM technology. Measurement of stress in SLM metal forming layer using different scanning strategies can provide guidance for the optimization of SLM process.

At present, residual stress measurement methods [7,8] can be divided into two categories: destructive methods (the small blind hole method [9], the stripping layer method [10], and the ring core method [11]) and nondestructive methods (the ray diffraction method [12], the magnetic memory method [13], and the ultrasonic method [14,15,16]). The destructive methods destroy the integrity of the component structure and can only realize the sampling testing. The detection depth of X-ray is very shallow, only 10–35 μm, mainly is used for measurement of stress in surface coating, film, etc. The magnetic memory method can only be used for measurement of ferromagnetic materials. Ultrasonic measurement method is the most potential stress testing method, which has the advantages of easy use, safety and on-line detections, therefore, this method is used to measure stress in SLM forming specimen.

Stress measurement by ultrasonic wave is based on acoustic-elastic theory [17,18]. At present, many research results [19,20,21] have been obtained in the stress measurement of materials with obvious acoustic-elastic effect (such as aluminum and its alloys, transparent glass for aviation, etc.), but there are few reports on the ultrasonic stress measurement of SLM forming parts. In this paper, SLM specimen manufactured with 316L stainless steel using different scanning strategies were as the test objects, the Critical Refraction Longitudinal (LCR) wave was used to measure stress [22], and the acoustic-elastic formula for measuring the stress in SLM forming specimen was established based on static load tensile test. The variation law of the acoustic-elastic curve was analyzed in combination with the microstructure of SLM forming specimen, and the measurement results were verified by X-ray and blind hole test.

## 2. Acoustic-Elastic Equation of LCR Wave

As shown in Figure 1, LCR wave is a special mode longitudinal wave which is generated when longitudinal wave incidents at the first critical angle *θ*_1_, it propagates below the medium surface in a direction parallel to the surface. The penetration depth of LCR wave is related to the ultrasonic excitation frequency [23]. The lower the frequency, the deeper the penetration depth is. The irregularity of propagation surface (such as slight crack) has little effect on the propagation of LCR wave. Therefore, LCR wave is the most sensitive method to measure surface stress compared with other wave types. The stress in SLM metal forming layer is plane stress [24]. It is assumed that the stress state in the measured micro-area is shown in Figure 2. Stress σ_1_, σ_2_ in the direction of X_1_ and X_2_ are evenly distributed in the measured micro-area. According to the theory of acoustic elasticity [25,26], the acoustic-elastic equation of LCR wave can be approximately expressed as:(1)$$\Delta v/v_{0}=(v_{0}-v_{i})/v_{0}=\sigma_{1}k_{i1}+\sigma_{2}k_{i2}$$
where $k_{i1}=\frac{\partial v_{i}}{\partial \sigma_{1}}|v_{i}=v_{0},\;k_{i2}=\frac{\partial v_{i}}{\partial \sigma_{2}}|v_{i}=v_{0}\;(i=1,2)$, *v*_0_ is the velocity of the LCR wave when the material under test is in stress-free state. The coefficients *k_i_*_1_, *k_i_*_2_ can be calibrated by measuring the change of LCR wave velocity when the material under test is in stress state.

## 3. Ultrasonic Measurement for Stress in SLM Metal Forming Layer

### 3.1. Experimental Materials and Specimen Preparation

The experimental material is 316L stainless steel spherical powder, the maximum particle size of the powder is 60 μm and the minimum particle size is 45 μm, apparent density is 4.42 g/cm^3^. The composition of the powder is shown in Table 1.

The used SLM equipment is AM250 (Reinshaw, Gloucestershire, UK) equipped with 200 W and 400 W lasers. Samples were built using layer thicknesses of 20 µm to 100 µm, and with a volume of 250 mm (X-axis) × 250 mm (Y-axis) × 300–360 mm (Z-axis). AM250 comes with a fully-welded vacuum chamber for low-pressure evacuation. The chamber keeps oxygen concentrations below 50 ppm through low gas consumption to enable safe use of reactive materials. According to the national standard GB/T228.1, smooth static load tensile specimens were manufactured by SLM technology. The size of the specimen is shown in Figure 3. Detailed processing parameters are shown in Table 2.

Meander, stripe and chess board scanning, respectively, were used in the experiments. The scanning path and sequence of the three scanning strategies are shown in Figure 4. The cross section is filled with scanning lines, and the divisional direction rotation mechanism are adopted in three strategies, the upper and lower scanning lines are staggered, the deflection angle of scanning lines belonging to the same area in adjacent layers is set to 67°.

The path planning of the meander scanning strategy is relatively simple, and the scanning paths in the same section are a group of parallel lines. When a scanning line is finished, the origin of the next adjacent scanning line will be next to the origin of the finished scanning line. The spacing between adjacent scanning lines should be less than the laser spot diameter. Compared with the other two scanning strategies, the scanning strategy is more efficient, but the scanning line is longer, which will produce greater accumulated stress in the forming parts. When forming small size parts and the quality requirement is not high, the meander scanning strategy can be used. 

In stripe scanning strategy, the section are divided into sub-areas by parallel lines with equal spacing, and allowing a small amount of overlap between the each two adjacent areas in the same layer. The overlap area is generally set to equal as a spot diameter. The direction of the scanning lines are perpendicular to that of the parallel lines. The length of the scanning lines are limited by the spacing of parallel lines. Compared with meander scanning strategy, the accumulated stress in forming parts is smaller. When forming medium size parts, stripe scanning strategy can be used.

Chess board scanning strategy is achieved by dividing the section into the isometrically square sub-areas, and allowing a small amount of overlap between two adjacent square areas in the same layer. This scanning strategy limit the length of the scanning lines inside each sub-area. The scanning lines follow the rules that the scanning lines in adjacent areas are orthogonal. The numbers in Figure 4 indicate the sequence of laser scanning. Compared with sequential scanning, this scanning sequence of alternate and intersect reduces the accumulation of thermal stress in the sub-areas. When forming large size parts, too long scanning line and excessive accumulated stress will have a negative impact on the quality of the parts. Therefore, when forming large size parts, chess board scanning strategy can be used to improve the quality of the forming parts.

### 3.2. Experimental Method

LCR wave stress measurement system is shown in Figure 5. It mainly consists of a Panametrics-NDT 5800PR (Panametrics-NDT, Waltham, MA, USA) ultrasonic pulse transmitting instrument, a TDS5000B oscilloscope (maximum sampling frequency 2.5 GHz, Tektronix, Beaverton, OR, USA), a transmitting transducer and a receiving transducer (SIUI, Shantou, China, with a frequency of 5 MHz). The distance between transducers is 20 mm. The electric pulse from the ultrasonic pulse transmitting instrument are outputted in two ways. One way to the piezoelectric chip of the transmitting transducer is used to excite the longitudinal wave signal. The other way goes through the oscilloscope synchronous trigger system to trigger the oscilloscope to collect the longitudinal signal received by the receiving transducer. The high-pass filter and low-pass filter of oscilloscope are used to filter the excitation and received signals. The cut-off frequency of high-pass filter is 2.5 MHz and that of low-pass filter is 7.5 MHz. The high-frequency interference signal of 7.5 MHz can be effectively filtered and the non-linear interference signal brought by the instrument can be eliminated. 

### 3.3. Experimental Data Acquisition Method

According to the acoustic-elastic Equation (1), it is necessary to measure the propagation velocities of LCR wave in specimen under different stress conditions. During the experiment, the velocity was indicated by measuring the propagation time of LCR wave within a fixed distance (20 mm). The LCR wave stress measurement system shown in Figure 5 was used to collect the LCR wave signal received by the transducer in the SLM tensile specimen after stress relief annealing. Measurement temperature is 26 °C. When the specimen is in stress-free state, the time *t_0_* of LCR wave propagation within 20 mm in the surface of the tensile specimen was recorded. The collected LCR wave signal is shown in Figure 6.

SLM tensile specimen was slowly loaded to the predetermined load at a loading speed of 1 kN/s on the WDW-200E (Youdao Material Testing Machine Co., Ltd, Jinan, China) static load tensile testing machine. When the load value of the testing machine was stable, a simple fixture was used to fix the ultrasonic transducers in the area to be measured to collect data. The load increment was 2 kN, five times of data were collected, respectively, along the directions parallel and perpendicular to the loading direction under each load, and the process was repeated until obvious plastic deformation or crack occurs in the specimen and the experiment was stopped. The stress-strain curves of the tensile specimens are shown in Figure 7. As can be seen from the Figure 7, the tensile strength and maximum strain of the specimen manufactured using chess board scanning is the highest, followed by stripe scanning and meander scanning. When fracture occurs, the maximum tensile strength of specimen manufactured using meander scanning is only 501.73 MPa and the maximum strain is only 17.07%, while the maximum tensile strength of specimen manufactured using stripe scanning is 598.6 MPa and the maximum strain is 28.04%, the maximum tensile strength of specimen manufactured using chess board scanning is 630.56 MPa and the maximum strain is 34.04%. Therefore, the scanning strategy can significantly affect the tensile strength of SLM forming parts when other processing parameters are the same.

## 4. Experimental Results and Analysis

### 4.1. Experimental Results

The reference signal *t_0_* is the LCR wave signal which was collected when the specimen was in stress-free state, and the calculation signal *t_i_* is the signal which was collected when the specimen was in stress state. The acoustic time difference Δ*t* of LCR wave signal propagating in a fixed sound path (20 mm) was calculated by cross correlation function [27] method. The calculation formula is shown in Equation (2), where m is the number of sampling points.
(2)$$\Delta t=\frac{m\sum_{n=1}^{n=m}t_{i}(n)t_{0}(n)-(\sum_{n=1}^{n=m}t_{i}(n))(\sum_{n=1}^{n=m}t_{0}(n))}{\sqrt{[m\sum_{n=1}^{n=m}t_{i}{}^{2}(n)-(\sum_{n=1}^{n=m}t_{0}(n){)}^{2}][m\sum_{n=1}^{n=m}t_{i}{}^{2}(n)-(\sum_{n=1}^{n=m}t_{0}(n){)}^{2}]}}.$$

The LCR wave acoustic-elastic curves parallel to and perpendicular to the direction of stress loading in tensile specimens were established. The results are shown in Figure 8. It can be seen from Figure 8 that the tensile stress is less than the critical stress value, the acoustic time difference of LCR wave increases linearly with the increase of stress, when the tensile stress is greater than the critical stress value (critical stress refers to the stress at the linear and non-linear transition points in the acoustic-elastic curve), the relationship between them no longer conforms to the linear variation law, but fluctuates up and down along the irregular curve. The critical stresses of the three specimens are 372 MPa (meander scanning), 465 MPa (stripe scanning), and 494 MPa (chess board scanning), respectively. The change rate of acoustic-elastic curve in the direction parallel to the loading direction is greater than that perpendicular to the loading direction.

### 4.2. Analysis of Experiment Results Based on Microstructure

Fracture morphology of specimen manufactured with SLM 316L stainless steel is shown in Figure 9. As can be seen from Figure 9, all the specimens show ductile fracture [27] characteristics on the whole, dimple characteristics can be seen from the fracture surface, and Specimen manufactured with SLM 316L stainless steel have good plastic deformation ability macroscopically. There are many micro-voids on the surface of the fracture. These voids grow up continuously under the action of stress and form dimples and cracks. 316L stainless steel has small crack propagation resistance and is very sensitive to cracks. Cracks accumulate to final fracture. Its micro-fracture is dimple-like, and its fracture mechanism is dimple-micropore aggregation. 

Figure 9 shows that SLM specimen manufactured with 316L stainless steel have good plastic deformation ability during static tension, and the deformation state of the material is the main factor that determining the change trend of acoustic time difference-stress curve of the LCR wave signal. When the tensile stress is less than critical stress, the material is in the stage of elastic deformation, so the acoustic time difference of LCR wave increases linearly with the increase of stress, which is consistent with the theory of acoustic elasticity of ultrasonic signals. When the stress is greater than critical stress, the acoustic-elastic curve no longer conforms to the linear variation law. It can be seen from Figure 7 that the macroscopic yielding stress of the three specimens are 469 MPa, 515 MPa, and 530 MPa, respectively. The critical stress is less than the macroscopic yielding stress. The reason is analyzed by combining the microstructure of the SLM 316L stainless steel forming layer shown in Figure 10. Because the specimens were manufactured by multi-channel overlapping and multi-layer stacking, there are obvious fusion interfaces in the SLM forming layer. In the same micro-molten pool, the grain boundaries are close, and columnar and cellular crystals grow preferentially along the direction perpendicular to the fusion interface or at a certain angle with the fusion interface. The columnar crystals and cellular crystals exhibit remarkable epitaxy growth characteristics between adjacent fusion interfaces. According to the above analysis results, the microstructure of SLM 316L stainless steel forming layer has obvious anisotropic characteristics. Therefore, when the tensile stress is larger than the critical stress, the internal deformation of each part of the specimen is uneven, the deformation of some zones (such as crystal boundaries, inclusions, fusion interfaces, etc.) have exceeded the elastic deformation range and entered the plastic deformation stage, while other zones (such as equiaxed crystal zones) are still in the elastic deformation stage [28]. The results of LCR wave stress measurement based on acoustic-elastic theory reflect the average stress of material in the range of ultrasonic propagation. The average deformation of specimen within the range of LCR wave measurement has entered the plastic deformation stage, the acoustic-elastic curve of LCR wave signal appears "fluctuation". The whole specimen has not entered the plastic deformation stage, with the increase of stress, the plastic deformation area gradually enlarges, when the tension stress is greater than macroscopic yielding stress, the whole specimen enter the plastic deformation stage until fracture. According to the above analysis, the inhomogeneous deformation of the anisotropic SLM forming specimen is the fundamental reason that the critical stress in the acoustic-elastic curve is less than the macroscopic yielding stress.

Combining with the fracture mechanism of dimple-micropore aggregation, the change rule of acoustic-elastic curve is analyzed theoretically. The experimental results show that SLM specimens manufactured with 316L stainless steel have good plastic deformation ability macroscopically. The theory of metal plastic deformation shows that the plastic deformation of metal material is the reflection of dislocation movement [29,30]. Dislocations are defects in crystals or imbalances in crystals. The dislocation will slip under the continuous external force. The dislocation is first initiated on the slippable surface. When dislocation slips on a slip surface, if it encounters obstacles (such as crystal boundaries, inclusions, fusion interfaces, etc.), the dislocation will be “blocked” and dislocation pile-up will be formed. Under the action of external force, the number of dislocations increases, the stress caused by the dislocation pile-up increases gradually. For SLM specimen manufactured with 316L stainless steel, the stress concentration caused by dislocation pile-up cannot be relaxed or released through the plastic deformation. When the stress is greater than the binding force of the obstacles to dislocation slip (crystal fracture strength). The microvoids will be formed in the location where the dislocations are accumulated, and the microvoids will grow up gradually, gather to form dimples and cracks. In the stage of plastic deformation, with the further increase of stress, the cracks formed in the SLM specimen manufactured with 316L stainless steel gradually expand and the size gradually enlarges. Based on the theory of crack propagation [31], crack instability propagation is a dynamic process, in the plastic deformation stage, when the external stress is less than the critical value of stress at the crack tip (which is related to the material type and crack size) [31], the crack is in steady state. With the increase of stress, the stress at the crack tip increases gradually, so the propagation speed of LCR wave in the specimen decreases gradually, the acoustic time difference of LCR wave signal increases. When the stress value is greater than critical value of stress at the crack tip, the crack is in the state of unstable propagation. With the propagation of cracks, the stress is released and the velocity of LCR wave propagation increases, the acoustic time difference of LCR wave signal decreases. Under the action of external force, the crack propagates to a new length value, the critical value of stress for which crack extension occurs increases, so the crack stops extension. With the increase of external force, when the stress is greater than the critical value of stress for which crack extension occurs, the cracks continues to destabilize and propagate in SLM forming layer. The concentration-release-concentration process occurs. Therefore, in the stage of plastic deformation, the acoustic time difference of LCR wave signal in the SLM forming layer fluctuates obviously with the change of stress. 

In order to verify the above analysis results, The surface of SLM specimen (meander scanning) with 20 MPa and 400 MPa stress state were detected by ultrasonic microscope VUE 250-P (OKOS, Manassas, VA, USA).The longitudinal wave with a center frequency of 100 MHz was used to detect the specimens at a sampling frequency of 110 MHz. Based on the loading mode of specimens in the tensile experiment, the SLM specimens were scanned globally along the path parallel to the loading direction. The scanning results are shown in Figure 11. 

Figure 11 shows that when the stress is 20 MPa, the imaging results of the SLM specimen manufactured with 316L stainless steel are basically uniform. When the stress is 400 MPa, the uneven “stripe” areas will appear in the imaging results of the specimen. According to the principle of ultrasonic microscopy [32,33], the bigger the difference of color brightness is, the bigger the defect size is. The imaging results show that there are many micro-cracks on the surface of the SLM specimen manufactured with 316L stainless steel, which is basically consistent with the results of theoretical analysis.

### 4.3. Linear Fitting Results of Acoustic-Elastic Curves

Based on the criterion of high correlation coefficient and small standard variance [34], the linear part of LCR wave acoustic-elastic curve in Figure 8 is fitted by least square method [35]. The fitting results are shown in Figure 12 and Equations (2)–(4).
(3)$$\left\{ \begin{array}{l} t_{1}-t_{1}^{0}=0.072\sigma_{1}+0.0465\sigma_{2}-2.41\\ t_{2}-t_{2}^{0}=0.0465\sigma_{1}+0.072\sigma_{2}-2.6 \end{array}\right.$$
(4)$$\left\{ \begin{array}{l} t_{1}-t_{1}^{0}=0.0511\sigma_{1}+0.033\sigma_{2}-1.7132\\ t_{2}-t_{2}^{0}=0.033\sigma_{1}+0.0511\sigma_{2}-1.55 \end{array}\right.$$
(5)$$\left\{ \begin{array}{l} t_{1}-t_{1}^{0}=0.0542\sigma_{1}+0.0361\sigma_{2}+2.7375\\ t_{2}-t_{2}^{0}=0.0361\sigma_{1}+0.0542\sigma_{2}+2.74 \end{array}\right.$$
where *t_i,_ t_i_*^0^ (*i* = 1, 2) is the time taken by LCR wave propagating the same distance along the measurement direction when the specimen is in stress-free and stress state respectively. Equations (3)–(5), respectively, are the acoustic-elastic formulas for measuring stress in SLM 316L stainless steel manufactured using meander, stripe and chess board scanning. According to the above experimental results, the microstructures, tensile strength, and yield strength of the SLM forming specimens vary with the scanning strategies. Therefore, there are differences in the acoustic-elastic formulas for measuring stress in specimens manufactured using different scanning strategies. The acoustic-elastic coefficients [36] parallel to the loading direction are 0.072 nsMPa^−1^, 0.0511 nsMPa^−1^, and 0.0542 nsMPa^−1^, respectively, perpendicular to the loading direction are 0.0465 nsMPa^−1^, 0.03311 nsMPa^−1^, and 0.036 nsMPa^−1^, respectively. When the scanning strategy is the same, the acoustic-elastic coefficient parallel to the loading direction are slightly larger than that perpendicular to the loading direction. The third term in the acoustic-elastic formula is a constant, which indicates the initial residual stress (residual stress after stress relief annealing) in the specimen when the specimen is in stress-free state.

## 5. Comparison of Different Stress Measurement Methods

Static tensile specimen 1 (meander scanning), specimen 2 (stripe scanning), and specimen 3 (chess board scanning) were manufactured by SLM technology. The distribution of residual stress measurement points in X direction on the surface of the parts were shown in Figure 13. Ten measuring points were evenly distributed along the center line of X direction on the plane, and the spacing of each measuring point is 15 mm.

Before measurement, it is necessary to smooth the surface of the part to be measured with sandpaper. The surface roughness Ra of the tested area should be less than 10 µm, the average surface roughness Ra of the specimens manufactured using three scanning strategies was 11.35 µm, so the tested area had been polished by p400 SiC sandpaper for about 20 minutes. And then remove the oil stain on the surface with acetone, so as to ensure good coupling between the LCR transducers and the part to be measured. TM-100 medical ultrasonic coupling agent was used for the ultrasonic inspection. Keep the measurement temperature constant (26 °C). The experimental results showed that the effect of surface treatment on residual stress measurement was less than 2%. The reason is that surface preparation only affects the residual stress in the range of 0.025 mm below the surface of the specimen. The measurement results of LCR wave reflect the average stress in the range of ultrasonic propagation (the penetration depth of LCR wave in 316L stainless steel is 1.2 mm). In practical application, when the surface roughness Ra of specimen is less than 10 μm, surface preparation is not performed before testing.

The used X-ray stress testing instrument is X-350A (GR/X-350A, Beijing, China). The surfaces of the specimens were electrolytically corroded by HCL + HNO_3_ (volume ratio 3:1). The corrosion depth was 0.6mm. Tilt fixed *ψ* (*ψ* = 45°) method was used, 2*θ* scanning range from 125° to 136°. The diffraction crystal plane (220) is irradiated by CrKa (λ = 229.1 pm). Measured the direction and size of diffraction peak displacement by angular intersection method.

The blind hole stress testing instrument used is HK21A (GR/X-350A, Jinan, China). The surface of the specimen and the positive and negative sides of the strain rosette were wiped with alcohol, and the wiping direction followed a single direction, as required. Strain rosette should be stuck immediately when the alcohol on the surface of specimen and strain rosette evaporated to dry. The drill axis was perpendicularly aligned with the strain rosette center, and the diameter of the hole was 1.5 mm, and depth was 0.6 mm. The instrument was turned on and set to zero. If the zero-setting number drifted evenly around zero, the operation was correct. When the measuring number was stable, the residual stress measurement was completed.

The residual stresses at each measuring point in Figure 13 were measured by LCR wave, X-ray, and blind hole methods, respectively. Figure 14, Figure 15 and Figure 16, respectively, are the results of residual stress measurement at each measuring point of the specimens using three measurement methods. It can be seen that ***σ_x_*** and ***σ_y_*** were all in tension stress state at the edge of the specimens, whatever scanning strategy was used. ***σ_x_*** and ***σ_y_*** were the largest when meander scanning strategy was adopted, while ***σ_x_*** and ***σ_y_*** were the smallest when chess board scanning strategy was adopted. The length of scanning line of chess board scanning was limited to each sub-area and scanning sequence of alternate and intersect was adopted, resulting in the reduction of thermal stress accumulation effect, and the residual stress in the SLM forming specimen reduced effectively.

As shown in Figure 14 and Figure 15, when meander scanning strategy was adopted, the distribution of ***σ_x_*** along X-direction fluctuated greater than that of ***σ_y_***, which varied from −172 Mpa to 50 Mpa. Due to the scanning line being long, and the laser beam scanned along the X-direction, result in the temperature gradient in the X-direction was greater than that in the Y-direction. So, the residual stresses in the SLM forming specimen fluctuated greatly. ***σ_x_*** and ***σ_y_*** in the same measure point of the specimen manufactured using stripe scanning strategy were smaller than that of specimen manufactured using meander scanning strategy. Because the length of the scanning lines was equal to the width of the sub-areas, it was much shorter than that using meander scanning strategy. So, the specimen manufactured using stripe scanning strategy presented a smaller temperature gradient, and the residual stress was also smaller. 

As shown in Figure 16, when chess board scanning strategy was adopted, the distribution of ***σ_x_*** and ***σ_y_*** along X-direction are basically identical, and the overall stress values were smaller than that using meander and stripe strategies. The distribution of ***σ_x_*** and ***σ_y_*** were more uniform. Because the length of scanning line of chess board scanning was limited to isometrically square sub-areas and scanning sequence of alternate and intersect was adopted, the stress in the SLM forming specimen was effectively reduced. It can be seen that the measured results of the three measurement methods are basically identical, and there are deviations in the stress values at the corresponding positions; the maximum deviation is about 6.2%, and the measurement error is within the allowable range.

## 6. Conclusions

(1) The acoustic time difference of LCR wave in SLM 316L stainless steel forming layer increases linearly with the increase of the stress, when the tensile stress is less than critical stress(372 MPa (meander scanning), 465 MPa(stripe scanning), and 494 MPa (chess board scanning)). The acoustic time difference of LCR wave in SLM 316L stainless steel forming layer no longer conforms to the linear law, showing a wave-like trend when the tensile stress is greater than critical stress. The analysis shows that the non-uniform deformation of the anisotropic SLM forming layer and the dimple-micropore aggregation fracture mechanism are the main reasons for the fluctuation of the acoustic-elastic curve in the plastic deformation stage. 

(2) The microstructures, tensile strength, and yield strength of the specimen manufactured with SLM 316L stainless steel vary with the scanning strategies. The quality descending order of SLM forming specimens using different scanning strategies is as follows: chess board scanning, stripe scanning, and meander scanning. Therefore, there are differences in the acoustic-elastic formulas for measuring stress in SLM 316L stainless steel forming parts manufactured using different scanning strategies.

(3) Nondestructive measurement of stress in SLM forming specimen can be realized by using the LCR wave method. The stresses were all in tension stress state at the edge of the specimens, whatever scanning strategy was used. Sub-area scanning and scanning sequence of alternate and intersect were adopted, which can effectively reduce the stress in the SLM forming specimen. The overall stress values of SLM forming specimen manufactured using chess board scanning strategy were smaller than that using meander and stripe strategies. The distribution of stress were more uniform. 

## Figures and Tables

**Figure 1 materials-12-02719-f001:**
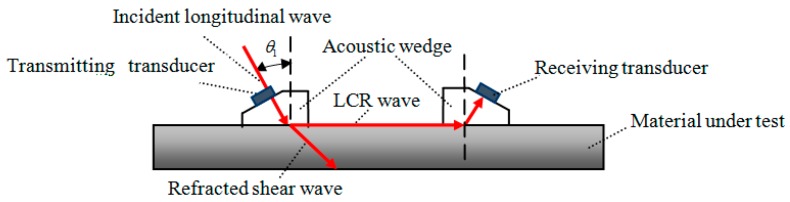
Schematic diagram of propagation of Critical Refraction Longitudinal (LCR) wave.

**Figure 2 materials-12-02719-f002:**
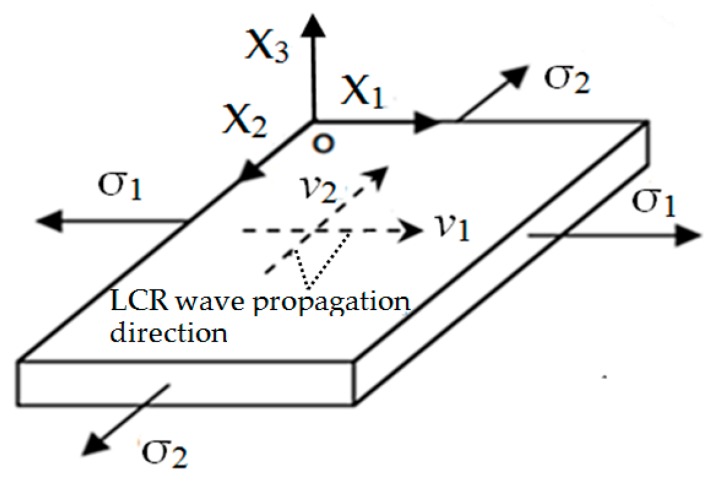
LCR wave vector in Selective Laser Melting (SLM) metal forming layer.

**Figure 3 materials-12-02719-f003:**
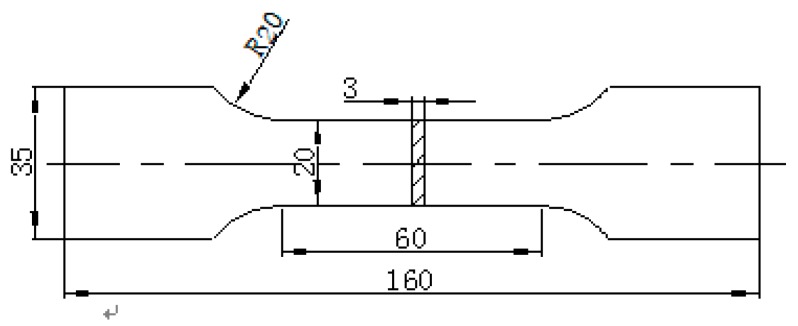
Schematic diagram of tensile specimen, size in mm (millimeter).

**Figure 4 materials-12-02719-f004:**
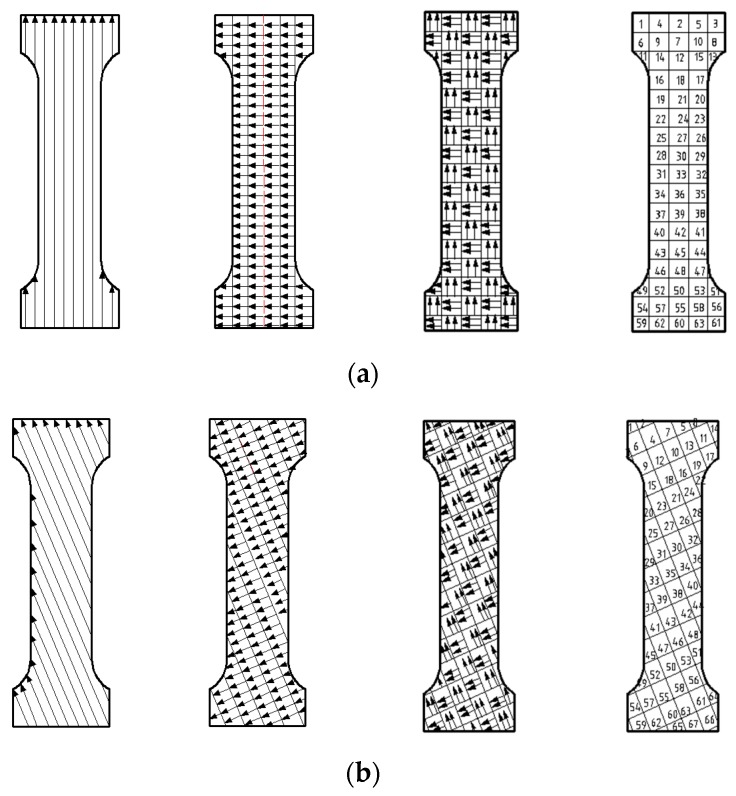
Schematic diagram of scanning path of meander scanning, stripe scanning, chess board scanning, and sequence of chess board scanning. (**a**) Scanning paths of layer N; (**b**) scanning paths of layer N + 1.

**Figure 5 materials-12-02719-f005:**
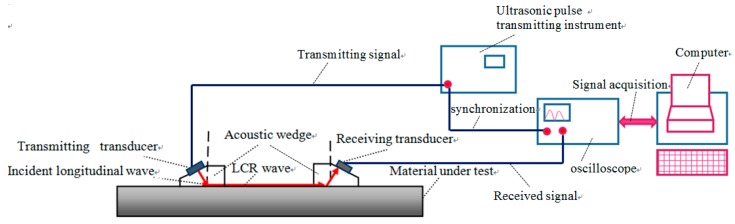
Schematic diagram of LCR wave stress measurement system.

**Figure 6 materials-12-02719-f006:**
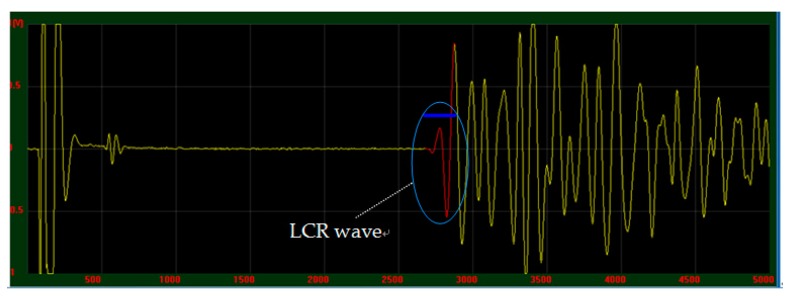
The LCR wave signal acquisition interface.

**Figure 7 materials-12-02719-f007:**
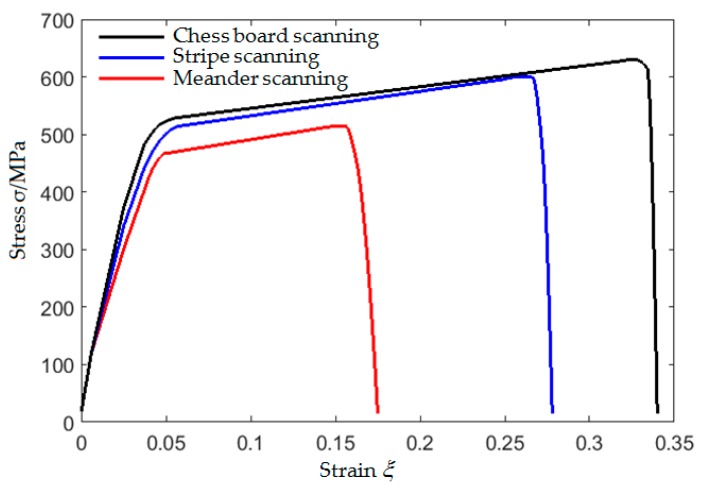
Stress-strain curves.

**Figure 8 materials-12-02719-f008:**
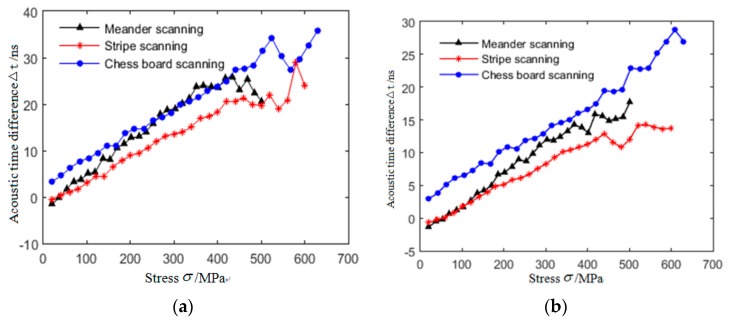
Acousto-elastic curves of LCR wave in tensile specimens manufactured by SLM technology under different scanning strategies. (**a**) Parallel to loading direction; (**b**) perpendicular to loading direction.

**Figure 9 materials-12-02719-f009:**
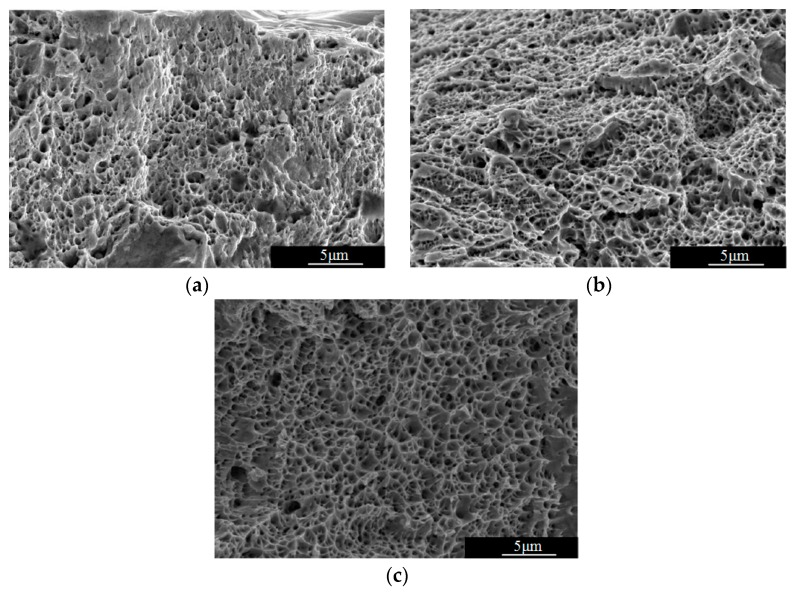
Fracture morphology of SLM specimen manufactured with 316L stainless steel. (**a**) Meander scanning; (**b**) Stripe scanning; (**c**) Chess board scanning.

**Figure 10 materials-12-02719-f010:**
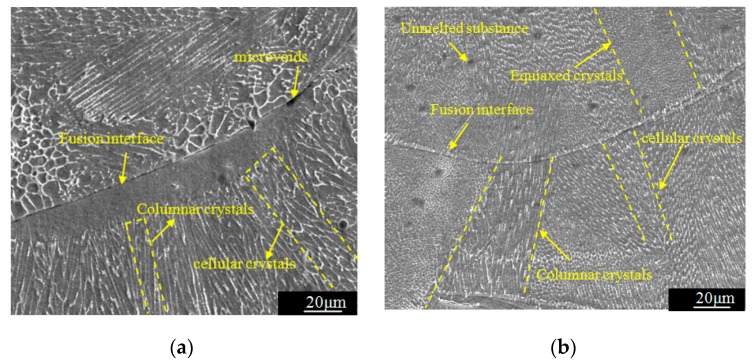
Microstructure of SLM specimen manufactured with 316L stainless steel. (**a**) Meander scanning; (**b**) stripe scanning; (**c**) chess board scanning.

**Figure 11 materials-12-02719-f011:**
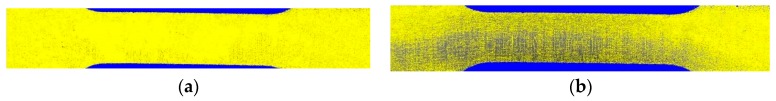
Imaging result of SLM specimen manufactured with 316L stainless steel. (**a**) Scanning imaging result of specimen with 20 MPa stress state; (**b**) scanning imaging result of specimen with 400 MPa stress state.

**Figure 12 materials-12-02719-f012:**
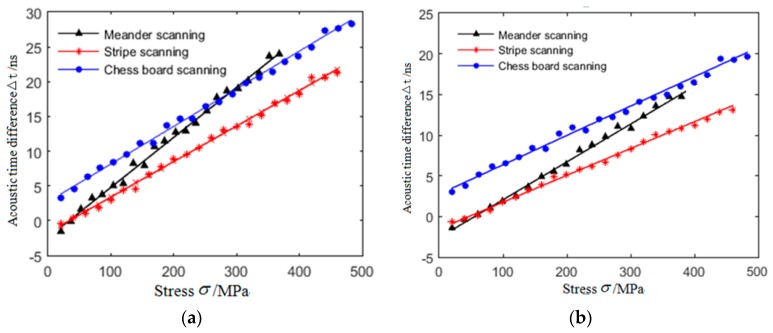
Linear fitting results of acoustoelastic curves of LCR wave. (**a**) Parallel to loading direction; (**b**) perpendicular to loading direction.

**Figure 13 materials-12-02719-f013:**
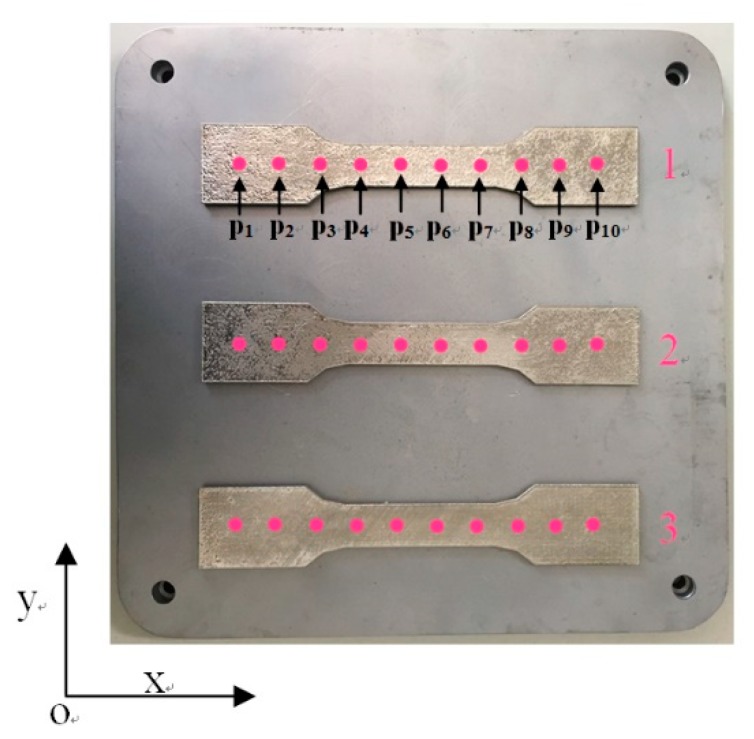
Schematic diagram of measurement points of residual stress.

**Figure 14 materials-12-02719-f014:**
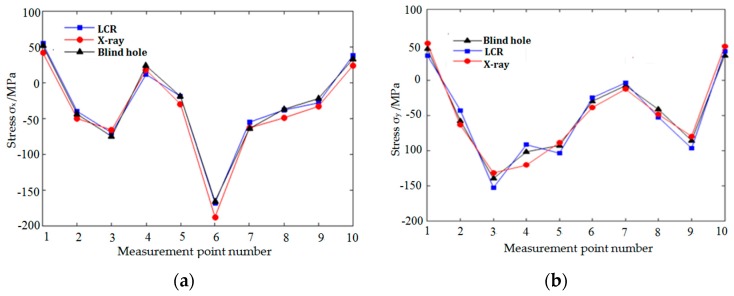
Residual stress distribution under meander scanning strategy. (**a**) The distribution of ***σ_x_*** along X-direction; (**b**) the distribution of ***σ_y_*** along X-direction.

**Figure 15 materials-12-02719-f015:**
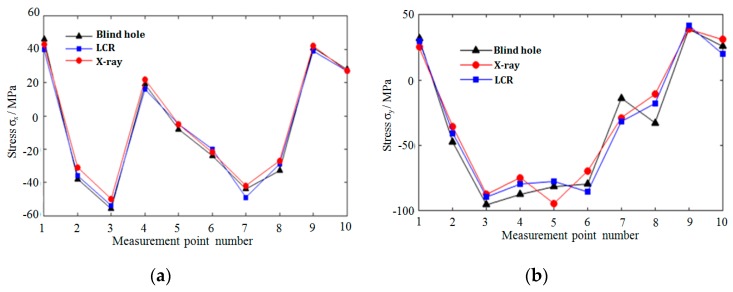
Residual stress distribution under stripe scanning strategy. (**a**) The distribution of ***σ_x_*** along X-direction; (**b**) the distribution of ***σ_y_*** along X-direction.

**Figure 16 materials-12-02719-f016:**
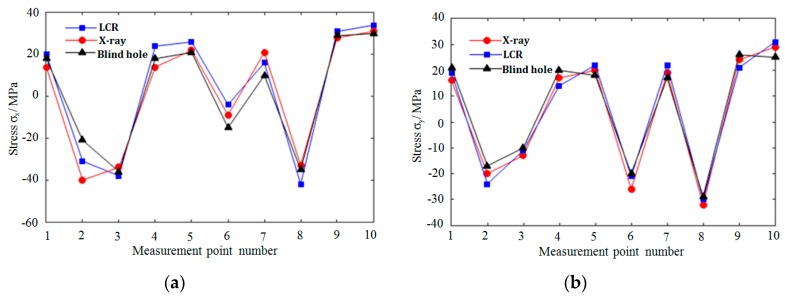
Residual stress distribution under chess board scanning strategy. (**a**) The distribution of ***σ_x_*** along X-direction; (**b**) the distribution of ***σ_y_*** along X-direction.

**Table 1 materials-12-02719-t001:** Chemical compositions of 316L stainless steel spherical powder (mass fraction, %).

Element	C	Cr	Ni	Mo	Si	Mn	O	P	Fe
Content	0.03	17.5	12.06	2.06	0.86	0.3	0.1	0.04	Bal.

**Table 2 materials-12-02719-t002:** Main processing parameters.

Laser Power (W)	Scanning Speed (mm/s)	Layer Thickness (μm)	Scanning Interval (mm)	Spot Diameter (μm)	Volume Fraction of Oxygen (%)
200	750	30	0.07	80	0.03

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
