# Peer review of "Ultrasonic Measurement of Stress in SLM 316L Stainless Steel Forming Parts Manufactured Using Different Scanning Strategies"

_materials, 2019, doi:10.3390/ma12172719_

Round 1

Reviewer 1 Report

2-4: Title need to be modified: "Ultrasonic Measurement of Stress in SLM 316L stainless steel Manufactured by Different Scanning Strategies"

The samples were not manufactured by the strategies, they were manufactured using or with different scanning strategies.

30-31: "Compared with traditional manufacturing technology, SLM technology 30 is theoretically not limited by any space, " What do you mean with any space? Maybe you wanted to say it is not limited to complex geometries

104-107: "SLM equipment adopts AM250( Reinshaw, Gloucestershire, England). AM250 Utilizing a high-powered fiber laser to print 20μm to 100μm thick layers of fully dense metal directly from 3D CAD data, With a build envelope of 250 mm (X-axis) x 250 mm (Y-axis) x 300-360 mm (Z-axis)."

This phrase is not well written, from English writing but also, concept-wise and punctuation. Example: The used SLM equipment is AM250 (Renishaw, Gloucestershire, England) equipped with a 200 W and  400 W lasers. Samples were built using layer thicknesses of 20 µm to 100 µm. ...

122: "The model size of the experimental processing is shown in Figure 3.” what do you mean here? You can already explained before the geometry of the samples

134: Figure 4: the font size is not consistent.

145: no comma, it should be a dot before the start of a sentence.

180-182: "Stop loading and do not unload, when the load of the testing machine is stable, a simple device with the same pressure was used to place the ultrasonic transducers in the area to be measured to collect data."

The first part of the sentence is not correct to use in as an order. Please rephrase it.

Etc.

General:

•After a comma always leave a space. After a parenthesis or bracket, introduce one space and continue with lower case.
•Please, correct the text thoroughly considering English, looking at sentence coherency, punctuation... It is not adequate to send to a journal in this form.
•Is it correct to call the samples as forming like "SLM 316L stainless steel forming specimens"? This term can be misunderstood with sheet metal forming process.

Author Response

Dear Reviewer:

 Thank you very much for the time and effort that you have put into reviewing the previous version of the manuscript. Your suggestions have enabled me to improve my work greatly. Based on your comment  and request, I have  made extensive  modification  on the original manuscript. Here, I attached revised manuscript in the formats of  MS word, for your approval. Appended to this letter is my point-by-point response to the comments raised by you.  The comments are reproduced and my responses are given directly afterward in a different color (red).

A revised manuscript with the correction sections red marked was attached as the supplemental material and for easy 

check/editing purpose.  

Should you have any questions, please contact me without hesitate. 

Kind regards

Yan XiaoLing

Reviewer 2 Report

Table 2 represents the SLM processing parameters. The data in the table should be represented uniformly in mm and not interchangeably - mm/um. Furthermore, it is not clear which data is the "scanning interval". It can be assumed that it is a spacing between two adjacent laser tracks, but given the spot diameter of 0,07mm and the scanning interval of 0,08mm, it means that there is no overlapping between the two adjacent tracks, which is rather strange and usually produces a porous part. These data should be explained in more details or corrected.

There is also a small typing mistake where POWER and POWDER are used in wrong places. This repeats throughout the text and should be improved.

The whole text should be proofread for grammar, spelling and writing style.

Author Response

(The authors gave the same response as above.)

Reviewer 3 Report

The Authors study the capability of ultrasonic methods based on Longitudinal Critical Refraction (LCR) waves for the measurement of stresses in Selective Laser Melting (SLM) obtained 316L stainless steel specimens. The Authors consider three different scanning strategies for SLM manufacturing: meander scanning, stripe scanning and chess board scanning. Experimental results show how the scanning strategy influence the acoustoelastic behavior; the latter analysis is supported by micromechanical considerations. Finally, the comparison with other experimental techniques for determining residual stresses allows for validating the proposed acoustoelastic approach.

The subject of the paper is very interesting and plays a crucial role in the greatly developing technology of additive manufacturing. The research is well developed and enriched by a lot of interesting consideration and comments. Anyway, in my opinion there are some weakness to be amended before publication. Below I point out some comments that can be useful for revise the paper.

1) English is quite sloppy and would highly benefit form a complete revision by a mother tongue reader. There are some improper terms like “acoustic elasticity” instead of “acoustoelasticity” and similar, or “Laser Selective Melting” instead of “Selective Laser Melting”. Moreover, there are also some typos; only to list a few, see the isolated “l” in line 53, the capital “T” after the comma in the last part of line 98, “accoutic” in line 208, and so on. Finally, in many places there are missing spaces or too much spaces in the text.

2) I think that the impact of the paper would be better if in the text and in the keywords words like “additive manufacturing” and “3D printing” were mentioned;

3) In the Abstract, the three scanning strategies (meander scanning, stripe scanning and chess board scanning) are mentioned already in lines 15-16, before their description (lines 21-22); probably the part between lines 18 and 22 should be anticipated before line 14, for avoiding the impression of going back and forth with same concepts.

4) At the end of Introduction, line 65, something more should be specified about “supporting test” (by the way, tests?).

5) In Section 2, the sentence in line 73 “The stress in SLM metal forming layer is plane stress” should be supported by a suitable reference. Moreover, it is not clear the meaning of the sentence “the stress in plane stress field can be decomposed into a group of vertical stresses” (lines 74-75).

6) In Section 3.1 (or in the Introduction) it would perhaps be better to say something more about the three scanning strategy considered and about their relevance in applications. Moreover, the number in Figure 4 are excessively small.

7) In Section 3.3 the sentence about the time-of-flight measurement (lines 168-171) should be definitely improved, as well as the sentence in lines 180-183; moreover in Figure 7 the arrows after the text should be removed.

8) In Section 4.1 the sentence “Where m is the number of sampling points” (line 213) should be moved after Eq. 2 is mentioned (line 210). Furthermore, a better and clearer correlation (with some in-depth comments) between the macroscopic yielding stress and the stress corresponding to the loss of the linearity of the acoustoelastic curves (called “critical stress”) should be made in the text: for example, for meander scanning according to Figure 7 the macroscopic yielding stress is about 470 MPa, whereas the critical stress is 372 MPa. Overcoming the latter issue would improve strongly the paper and would create a better link with the micromechanical considerations.

9) The title of Section 4.2 should be improved, for giving the clear understanding that the experimental results discussed therein refer to micromechanical aspects (and not to LCR ultrasonic waves stress measurement). The text of the whole Section is a little bit hard to read, and sometimes appears to be repeating. Since I’m in Continuum Mechanics and Mechanics of Material, the concept of “critical stress” appears to create confusion, and appears to be not well related to the yield stress (see also comment 8)). Finally, I think that the claim “When the stress is greater than critical stress … the acoustic-elastic theory is no longer applicable to the deformation stage” (lines 286-289) it is not correct: indeed, this happens because the acoustoelastic theory considered in the MS is the classical one by Hughes & Kelly (https://doi.org/10.1103/PhysRev.92.1145, I think that this paper has to be listed in the references), that assumes that the stressed state is reached through a finite elastic deformation, but there are acoustoelastic theories that does not need hypothesis on the nature of the phenomenon generating the stress.

10) In Section 4.3, line 358, I guess that LCR waves have to be mentioned instead of Rayleigh waves.

11) In Section 5, something more about X-ray method and blind hole method should be specified. Moreover, about the distribution of residual stresses in the length of the specimens, whereas meander scanning (Figure 14) and stripe scanning (Figure 15) show similar behavior (but some differences are apparent anyhow), chess board scanning (Figure 16) shows a completely different behavior: thus, the description of these distribution cannot be unified as the Authors do.

12) In the Conclusions, point (3), the same considerations about the differences between the three examined scanning strategies in the distribution of residual stresses contained in comment 11) hold.

13) In the References there are some typos like, e.g., “modeling” instead of “Modeling” in [3], “Johson” instead of “Johnson” in [17]; moreover, reference [23] is the same of reference [18]. Finally, I suggest of improving references by listing (and cite suitably in the text) more papers relevant for the issues covered by the MS:

-for what concerns acoustoelasticity, add the paper cited in comment 9) and some other classical papers in the field

for what concerns the  determination of the acoustoelastic coefficients and the discussion of the variation of acoustoelastic coefficients with the stress in the Hughes & Kelly acoustoelastic theory, the paper https://doi.org/10.1016/j.proeng.2017.09.494;

- for what concerns LCR ultrasonic waves method for evaluating residual stresses, relevant papers like, e.g., https://doi.org/10.1016/j.apacoust.2018.07.017 etc.

Author Response

Dear Reviewer:

 Thank you very much for the time and effort that you have put into reviewing the previous version of the manuscript. Your suggestions have enabled me to improve my work greatly. Based on your comment  and request, I have  made extensive  modification  on the original manuscript. Here, I attached revised manuscript in the formats of  MS word, for your approval. Appended to this letter is my point-by-point response to the comments raised by you.  The comments are reproduced and our responses are given directly afterward in a different color (red).

A revised manuscript with the correction sections red marked was attached asthe supplemental material and for easy 

check/editing purpose.  

 Should you have any questions, please contact me without hesitate. 

Kind regards

Yan XiaoLing

Reviewer 4 Report

In the presented manuscript, an ultrasonic stress measurement technique  using LCR wave, is utilised to test its applicability for SLM-scanned 316L stainless steel. Three different scanning strategies i.e. meander scanning, stripe scanning and chess board scanning, were used as benchmarks. The paper is clearly written (except from some minor language errors) and it reads well. Authors provide a short but yet comprehensive introduction, then describe the experiment and testing procedure. Results are explained with the aid of SEM micrographs, where fundamental fracture mechanisms are identified and related to the main discussion. Conclusions are also clear.

In general, the paper shows interesting results of utilisation of nondesctructive method of residual stress assessment in 3D printed steel specimen. In my opinion it can be published after some minor corrections have been made:

1) There are some language issues - please proof-read the manuscript and correct them (e.g. line 186 "... strain of the specimen manufactured by chess board scanning is highest" ... should be "the highest" etc...

2) page 5, line 180: Please rewrite the sentence: "Stop loading and do not unload, when the load of the testing machine is stable, a simple device with the same pressure was used to place the ultrasonic transducers in the area to be measured to collect data." 

It does not read well.

3) Please explain what contact medium was used for the ultrasonic inspection. More importantly please provide more details on sample preparation (for the ultrasonic measurements). What was the gradation of the SiC paper used and how long the polishing was performed for? Please comment on the possible effect of the surface preparation on the residual stress level?

4) What Authors mean by critical stress (page 6. line 217)? Is it yield stress? Please explain.

Author Response

(The authors gave the same response as above.)

Round 2

Reviewer 3 Report

The Authors revised the paper according to the Reviewer's comments. The presentation has been substantially improved, although the English can be still enhanced.